ecology/cognition/behaviour

animal behaviour, cognition, contest, crustacean, pollutants, synthetic polymers

**Author for correspondence:**
Eoghan M. Cunningham
e-mail: ecunningham18@qub.ac.uk

# Animal contests and microplastics: evidence of disrupted behaviour in hermit crabs *Pagurus bernhardus*

Eoghan M. Cunningham[1,2], Amy Mundye[2],
Louise Kregting[1,3], Jaimie T. A. Dick[1,2],
Andrew Crump[2,4], Gillian Riddell[2] and Gareth Arnott[2]

[1]Queen's University Marine Laboratory, Queen's University Belfast, 12–13 The Strand, Portaferry BT22 1PF, Northern Ireland, UK
[2]Institute for Global Food Security, School of Biological Sciences, Queen's University, Belfast BT9 5DL, Northern Ireland, UK
[3]School of Natural and Built Environment, Queen's University Belfast, Belfast BT9 5BN, Northern Ireland, UK
[4]Centre for Philosophy of Natural and Social Science, London School of Economics and Political Science, London WC2A 2AE, UK

EMC, 0000-0002-6323-079X; LK, 0000-0001-7217-0146;
AC, 0000-0003-4485-5740; GA, 0000-0003-1525-3708

Microplastics are ubiquitous in global marine systems and may have negative impacts on a vast range of species. Recently, microplastics were shown to impair shell selection assessments in hermit crabs, an essential behaviour for their survival. Hermit crabs also engage in 'rapping' contests over shells, based on cognitive assessments of shell quality and opponent fighting ability and, hence, are a useful model species for examining the effects of microplastics on fitness-relevant behaviour in marine systems. Here, we investigated how a 5-day microplastic exposure (25 microplastics/litre) affected the dynamics and outcome of 120 staged hermit crab contests. Using a 2 × 2 factorial design, we examined how microplastics (i.e. presence or absence) and contestant role (i.e. attacker or defender) affected various behavioural variables. Significantly higher raps per bout were needed to evict microplastic-treated defenders when attackers were pre-exposed to control conditions (i.e. no plastic). Also, significantly longer durations of rapping bouts were needed to evict control-treated defenders when attackers were pre-exposed to microplastics. We suggest that microplastics impaired defenders' ability to identify resource holding potential and also affected attackers' rapping strength and intensity during contests. These impacts on animal contests indicate that microplastics have broader deleterious effects on marine biota than currently recognized.

# 1. Introduction

The input of plastic waste to the oceans shows no sign of slowing, with global production levels now exceeding 350 million tonnes per annum, and trending to reach approximately 700 million tonnes per annum by 2040 [1,2]. Microplastic (particles less than 5 mm) pollution is ubiquitous, from tropical coastal systems [3] to the depths of the Arctic Ocean [4]. Due to their small size, microplastics are more bioavailable to a greater number of marine biota than larger macro-sized plastic items, with ingestion documented in a range of animals from microscopic zooplankton [5] to giant baleen whales [6]. Microplastic pollution is a major threat to marine ecosystems [7], as they are known to enter and pass through food webs [8], cause false satiation through retention in the digestive tract of animals [9], translocate into vital tissue and organs [10] and act as a vector for the transport of endocrine-disrupting compounds [11] which are known to bioaccumulate in top predators [12].

The impacts of microplastics on animals vary depending on a number of factors, such as particle dosage, size, shape and polymer type [13], although polyethylene is the most ingested microplastic particle in marine systems [14]. The physiological impacts of microplastics on marine fauna [15] range from increasing oxidative stress (e.g. *Dicentrarchus labrax*; [16]) to reducing reproductive output (e.g. *Emerita analoga*; [17]). However, while physiological studies have been extensively reported in the literature [15], the impacts of microplastics on animal behaviour remain less understood. Previous studies have demonstrated that microplastics can affect social behaviours in fish [18,19], and also locomotion in bivalves [20], amphipods [21] and copepods [22]. Additionally, microplastics have also been shown to impact feeding behaviour in amphipods [23] and copepods [24], as well as coral species [25]. By contrast, microplastics have also been shown to have no impact on the feeding behaviour of shore crabs [26], which highlights that the effects of microplastics can differ among species. Thus, we require bespoke studies to examine the ubiquity of effects of microplastics on key survival traits across taxa.

Hermit crabs, a widespread marine crustacean, have a soft abdomen that requires external protection from predators, usually provided in the form of empty gastropod shells [27]. The number of empty gastropod shells in the environment is a limiting factor to hermit crab population size, highlighting the importance of this resource [28]. To attain a gastropod shell, hermit crabs can either enter an unoccupied shell or, more commonly, evict a conspecific and enter their vacated shell [29]. Hermit crabs gather information by investigating a new shell, which allows them to decide whether it is of a higher or lower quality than the shell they currently inhabit [30–32]. In this way, they offer a useful system for studying cognition, comprising the acquisition, processing, retention and use of information from the environment [33], and the effects of microplastics on key survival behaviours reliant on such information processing.

In particular, hermit crabs determine whether occupied shells are worth the energetic cost of a contest with their current inhabitant [34]. Larger hermit crabs generally assume the role of attacker during a contest and make contact with the defender's shell [35]. The defender will then withdraw inside its shell, as the attacker assesses the external surface for information on shell size and quality [30]. Defenders in the contest will assess the resource holding potential of an attacker (i.e. size, physiological state, weaponry; [36]) and determine whether contest costs (i.e. energy, injury and time) outweigh the quality of their own shell (i.e. resource value; [37]). The defenders do not perform any overt activities that the attackers can assess, so instead the attackers perform a self-assessment strategy based on physiological condition and levels of fatigue [36]. If either individual decides that its own resource holding potential is inferior, it will end the contest immediately unless the resource value is high enough [37]. If the attacker decides that shell quality outweighs contest costs, it will strike its own shell against the defender's shell [31]. This 'rapping' behaviour occurs in distinct bouts and will continue until either the defender evacuates the shell, termed an eviction, or the attacker retreats from the contest.

A recent study demonstrated that the time taken for common European hermit crabs (*Pagurus bernhardus*) to contact and subsequently enter an optimal-sized shell (i.e. shell selection) was significantly increased when exposed to polyethylene microplastics [38]. Given the importance of shell selection for survival, growth and reproduction, this is an essential behaviour required by this and other hermit crab species in order to attain suitably sized gastropod shells [38]. Additionally, microplastic exposure was shown to decrease startle response time in *P. bernhardus*, which also increases susceptibility to predation [39]. Hermit crabs inhabit the littoral zone, where microplastic pollution is found in high abundances [40], putting them at a higher risk of microplastic exposure. Given microplastic exposure affected the information gathering and decision making during shell selection behaviour in hermit crabs [38], there is a strong rationale that shell contest behaviour will

also be affected. The importance of investigating this is highlighted by shell contests being a major mechanism by which hermit crabs acquire shells, the size of which relative to the individual is a key determinant of fitness. Therefore, any impairment in this behaviour associated with microplastic exposure could have major consequences for environmental fitness, as lower quality shells can reduce growth, fecundity and survival [34]. In the present study, using a factorial experimental design, we investigated how microplastic presence affects contest dynamics and overall outcome in hermit crabs. We hypothesized that the decision making of hermit crabs during contests would be affected by the presence/absence of microplastic pollution, and/or the contestant role. The latter is important because the roles of attacker and defender are very different and distinctive in hermit crab contests, with attackers being highly active and using bouts of shell rapping to try to evict the defender who withdraws inside their shell and attempt to resist eviction. Given this, it is therefore important to investigate if microplastic exposure influences one contestant role more than the other. Additionally, we predicted that the overall outcome of hermit crab contests (i.e. the number of shell fights, evictions and swaps overall) would be affected by the presence/absence of microplastic pollution, and/or the contestant role.

# 2. Methodology

## 2.1. Collection and maintenance of animals

*Pagurus bernhardus* were collected at low tide from Ballywalter Beach, Co. Down, Northern Ireland (54.5408° N, 5.4840° W) between September and December 2020. The crabs were transported to Queen's University Belfast and maintained in a constant-temperature laboratory at 12°C with a 12 : 12 h L : D cycle. For three-week periods, hermit crabs were kept in glass stock tanks ($25 \times 25 \times 45$ cm) containing sand-filtered and constantly aerated seawater at a density of approximately 50 crabs per tank. All seawater was collected from the flow-through seawater system at Queen's University Marine Laboratory, Portaferry. To provide shelter, approximately 100 g of brown seaweed *Fucus serratus* was added to each tank and weekly water and seaweed changes were made. All hermit crabs were fed ad libitum on commercial fish food (catfish pellets) and returned to the shore at the end of the study.

## 2.2. Microplastic preparation

To test the effects of microplastic exposure on hermit crab behaviour, microplastic treatment tanks containing 10 l of sand-filtered and constantly aerated seawater and 50 g of polyethylene spheres were prepared (Materialix Ltd, London, UK; size: 4 mm, 0.02 g; [38]). The densities for each treatment were no added microplastics (i.e. control) or 25 microplastics per litre (MP l$^{-1}$) [38]. Microplastic densities were chosen based on low environmental values found throughout the literature [41]. Two tanks for both treatment groups ($n = 4$) were used to avoid any bias as a result of tank effects (e.g. water quality) and 20 crabs were allocated to each treatment tank using a random number generator and exposed for 5 days. Aeration was used to aid the distribution of microplastic particles, and each tank contained 80 g of the brown seaweed *F. serratus*. Polyethylene was chosen for this study as it is the most produced form of plastic in Europe and is ingested by marine organisms more than any other type of plastic [14].

## 2.3. Experimental design

This study had a $2 \times 2$ factorial design (i.e. two factors, each with two levels). The first factor, 'microplastics', described either the presence or absence of plastic particles. The second factor, 'contestant role', described whether the focal hermit crab was allocated to assume either the role of attacker (the larger individual of a contesting pair) or defender (the smaller individual of a contesting pair) in staged contests. Thus, there were four contest types: (i) both contestants from the control treatment (C); (ii) both contestants from the microplastic treatment (MP); (iii) the attacker from the MP and the defender from the C (MPA) and (iv) the attacker from the C and the defender from the MP (MPD). Each treatment was replicated 30 times (i.e. 120 contests in total). This design allowed us to test for interaction effects and whether microplastics impacted attacking or defending behaviour during the contests. For consistency and comparability, only male hermit crabs were used to avoid

any effects of sex and reproductive status on the behavioural observations [30]. Also, only crabs that were free of parasites and had both chelipeds intact were used in the experiments.

## 2.4. Shell allocation

At the end of the exposure period, a small bench vice was used to crack the hermit crabs out of their original shells (see [42]). No crabs were injured while being removed from their shell. The crabs were then dried with paper towels, weighed and sexed using a light microscope [30]. When a pair of males were identified, the larger hermit crab was allocated to a suboptimal *Littorina obtusata* shell (approx. 50% optimal weight; [43]), and the smaller hermit crab was allocated to the optimal shell weight for the larger crab (100%), which created an asymmetrical motivation for a 'shell fight' [31]. In addition, the relative weight difference (RWD) of the pair was calculated by the following equation: RWD = 1 − (small crab weight/large crab weight). Optimal shell weights were calculated using regression lines determined for preferred shell weight over a range of crab weights (y = 0.60 + 2.27x; [43]). The crabs were then left to acclimate in their given shells for 2 h in separate glass crystallizing dishes containing sand-filtered and constantly aerated seawater, giving them time to recover from handling and to assess the quality of their new shell. No animals were used more than once during the experiments.

## 2.5. Contest behaviour

After 2 h, both hermit crabs were placed in a 9 cm diameter crystallizing dish containing seawater to a depth of 7.5 cm, within a fully enclosed observation box containing two-way mirrors and a light to ensure the observer could not be seen during the contest. The subsequent contests were recorded using a Panasonic HC-V260 camera. The contests ended either when a crab was evicted from their shell and a shell swap was made or, if no eviction or shell swap occurred within 30 min, the contest was ended and recorded as a 'no contest'. The measured categorical variables (Yes/No) were contests that included a shell fight, contests that included an eviction, and contests that included a shell swap. Numerical variables measured were latency to contact the defender's shell (s), the number of bouts of rapping, the number of raps within a single bout, the duration of rapping bouts (s), the eviction time (s), the time taken for the attacker to swap and remain in the defender's shell (s), and the relative weight difference of the pairs.

## 2.6. Statistical analysis

All statistical analyses were carried out in R v. 3.4.4 [44]. The differences among the binary categorical variables were analysed using chi-squared tests ($\chi^2$). The numerical data were assessed for normality of residual distributions (Shapiro–Wilk test, $p > 0.05$) and homogeneity of variances (Fligner–Killeen, $p > 0.05$). All numerical data followed a non-normal distribution and homogeneity of variance and was therefore transformed to assume a normal distribution using a $\log_{10}$ transformation. A two-factor ANOVA was used to determine any significant differences in the behaviours displayed under the presence/absence of plastic (factor 1), when the crab was assuming the role of attacker/defender (factor 2), and also between the interaction of both factors [31]. In the case of a significant interaction effect, a *post hoc* Tukey's pairwise comparison test was used to highlight the factor levels that differed significantly from each other.

## 3. Results

No mortality was observed as a result of microplastic exposure throughout the experiment. Among all treatments overall, 67–77% of contests included shell fights, 67–73% included shell evictions and 57–70% included shell swaps (table 1). The overall outcome of hermit crab contests was not affected by the presence or absence of microplastics or the contestant role, as there was no significant difference in the percentage of contests that ended in shell fights ($\chi_4^2 = 1.18$, $p = 0.88$), shell evictions ($\chi_4^2 = 0.77$, $p = 0.94$) or shell swaps ($\chi_4^2 = 3.41$, $p = 0.90$; table 1).

However, a significant microplastic exposure × contestant role interaction was found for the 'number of raps within a bout' ($F_1 = 5.88$, $p = 0.02$). Despite the *post hoc* analysis reporting that no individual factor level differed significantly from each other, the significant interaction was driven by a higher average increase in the number of raps needed by attackers from control conditions to evict microplastic-

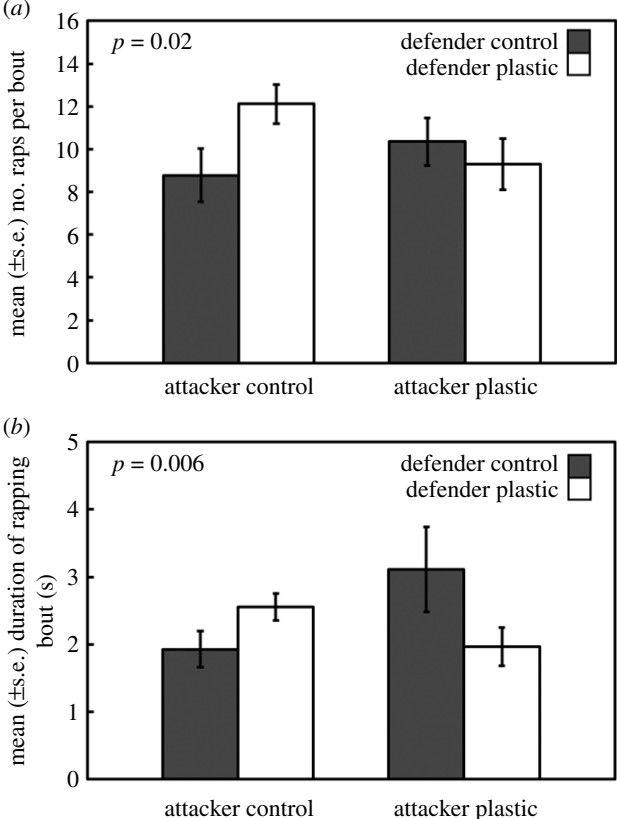

**Figure 1.** The mean (±s.e.) number of raps within a bout (*a*) and duration of rapping bouts (*b*) for each of the four treatment groups (C, MPD, MPA and MP). *p*-values refer to the interaction effect between microplastic (presence/absence) and the contest role (attacker/defender).

**Table 1.** The number and percentage of contests that contained a shell fight, shell eviction or shell swap overall for each of the four treatments; control (C), microplastic (MP), microplastic attacker (MPA) and microplastic defender (MPD).

| treatment | shell fight (% fights) | shell eviction (% evictions) | shell swap (% swaps) |
|---|---|---|---|
| C (*n* = 30) | 23 (77%) | 20 (67%) | 17 (57%) |
| MP (*n* = 30) | 21 (70%) | 21 (70%) | 20 (67%) |
| MPA (*n* = 30) | 22 (73%) | 22 (73%) | 21 (70%) |
| MPD (*n* = 30) | 20 (67%) | 21 (70%) | 20 (67%) |

exposed defenders from their shells (MPD; figure 1*a*). This difference was not apparent when the attacker was from the plastic treatment. Additionally, a highly significant microplastic exposure × contestant role interaction was found for the 'duration of rapping bouts' ($F_1 = 7.73$, $p = 0.006$). Again, despite a non-significant *post hoc* analysis, the highly significant interaction was driven by longer average durations of rapping bouts needed by attackers from microplastic conditions to evict control-exposed defenders from their shell (MPA; figure 1*b*). Furthermore, all of the other variables measured with numerical data were statistically non-significant (see electronic supplementary material, table S1).

## 4. Discussion

Microplastic exposure significantly affected hermit crab contest behaviour, both for defenders and attackers; however, this did not affect the overall outcome of the contests. Similarly, when Arnott & Elwood [31] manipulated resource value during hermit crab contests, there were effects on behaviour but not outcome, highlighting the need and utility of examining behaviour in detail. When in the attacker role, hermit crabs from control conditions used more raps per bout to evict defenders pre-

exposed to microplastics (MPD). This suggests that the ability of hermit crabs to defend their shell when pre-exposed to microplastics was increased when facing control attackers. Although this increase in defence did not affect the outcome of the contest, as the frequency of shell evictions and shell swaps were the same among treatments. Furthermore, attacking hermit crabs pre-exposed to microplastics used longer durations of rapping bouts to evict crabs from control conditions. The number of raps was not higher despite the increased duration of bouts, which suggests that the attacking ability of hermit crabs pre-exposed to microplastics was less effective or slower when attacking control defenders. We show that microplastics affect information gathering during these behaviours; however, further research is needed to see if microplastics directly affect hermit crab cognition.

Despite the level of resistance of microplastic-exposed defenders increasing when facing control attackers, the mechanisms behind this is unclear. It is possible that these defenders were impaired in their ability to accurately assess the resource holding potential of the attackers [36,45] and therefore defended their shells for longer than was needed. Furthermore, it is possible that the microplastic-exposed defenders were less likely to accurately assess the resource value of the shells they were residing in [37]. It may also be the case that the defenders exposed to microplastics became more risk-averse and decided to stay in their shells for longer despite facing an inevitable eviction. This increased investment in resistance suggests that the decision-making abilities of the microplastic-exposed defenders were impaired and is consistent with previous studies showing that microplastic impaired hermit crab decision making during other survival behaviours [38,39].

The mechanisms behind the reduced attacking ability of microplastic-exposed attackers when facing control defenders is most likely a result of the attackers' reduced physiological condition [46], as fighting ability largely depends on physiological condition of the animal [45]. Previous studies have shown how the rapping vigour of an attacker can be reduced as lactate levels increase throughout a contest [47]. Here, we highlighted longer durations of rapping bouts without having an increased number of raps, which suggests that the rapping behaviour was either slower or weaker. It is therefore possible that the 5-day exposure to microplastics affected the competitive ability of the hermit crabs. As the microplastics floated on the surface of the water and no microplastic ingestion was observed by the hermit crabs during the experiment, it is possible that chemical leachate from the polyethylene particles was having an effect [48]. Plastic products are made from polymers mixed with a blend of substances which are added to alter and enhance the properties of the plastic [49], including plasticizers (used to increase flexibility) and flame retardants (added to prevent ignition; [50]). In total, more than 6000 different additives are used in plastic production, many of which have detrimental effects on organisms [49]. In particular, polyethylene leachate was shown to affect the physiology of bivalve larvae after a 24 h exposure (*Meretrix meretrix*; [51]) by reducing shell heights and survivability in comparison to control treatments. Also, the settlement behaviour of barnacle larvae was also significantly reduced in the presence of polyethylene leachate over a 4-day period (*Amphibalanus amphitrite*; [52]), which highlights that leachate could be having negative effects on both the physiology and behaviour of the hermit crabs in our study. Other leachates, such as those from discarded cigarette butts, which are a prevalent form of plastic pollution in rocky shore habitats, have also shown to impact upon the physiology of mussels by reducing their filtration rates after a 5-day exposure [53]. Additionally, we used similar numbers of microplastics that have been found in environmental samples in the past [38,41] and, therefore, our density was not unrealistic. Future hermit crab studies would benefit from using separate leachate treatments alongside microplastic and/or nanoplastic particles to further investigate the effects of leachate on animal behaviour.

In conclusion, the attacking and defending ability of hermit crabs during shell contests were significantly affected by microplastic exposure. Our study is the first to assess the impact of microplastic on animal contest behaviour, adding to the mounting evidence that microplastics are influencing animal behaviour, including hermit crabs, which are an emerging model marine invertebrate [54]. Our results may also be important to understand how microplastics affect other crustaceans, particularly during contest behaviours when competing for other resources, such as burrows (*Nephrops norvegicus*; [55]) and food (*Carcinus maenas*; [56]). However, further research is needed to confirm how microplastics and chemical leachate affect the cognition of hermit crabs with focus needed on assessments of resource value and resource holding potential during contest behaviour. Determining the effects of microplastics and leachate on the physiology of hermit crabs, particularly lactate and glucose levels [46], would also help to determine how plastic pollution affects fighting ability. In terms of hermit crabs in the environment, we highlight potentially damaging effects of microplastics and leachate on survivability of this species, particularly as they reside in habitats such as rocky shores and the littoral zone that can be heavily polluted [40]. Furthermore,

future studies should also assess the reproductive output of hermit crabs under microplastic and leachate conditions to investigate how these pollutants may impact upon environmental fitness.

Ethics. Crustacean research is not regulated under UK law, but we followed the Association for the Study of Animal Behaviour's Guidelines for the Use of Animals in Research. After the experiment, all hermit crabs were returned to the shore unharmed.

Data accessibility. The data are provided in the electronic supplementary material [57].

Authors' contributions. E.M.C. and G.A. designed the study. E.M.C. and A.M. collected and maintained the animals in the laboratory and carried out the behavioural observations. E.M.C., A.M. and G.A. conducted the analysis. E.M.C. prepared the manuscript. All authors revised the manuscript, approved the final version and accept responsibility for its contents.

Competing interests. We declare we have no competing interests.

Funding. E.M.C. is supported by the Department for Agriculture, Environment, and Rural Affairs, Northern Ireland.

Acknowledgements. We thank the two anonymous reviewers whose comments greatly improved the manuscript.

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
