## [Peer Review File · Royal Society Open Science]

Review History

RSOS-211089.R0 (Original submission)

Review form: Reviewer 1

Is the manuscript scientifically sound in its present form?

Yes

Are the interpretations and conclusions justified by the results?

Yes

Is the language acceptable?

Yes

Do you have any ethical concerns with this paper?

No

Have you any concerns about statistical analyses in this paper?

Yes

Recommendation?

Major revision is needed (please make suggestions in comments)

Comments to the Author(s)

In this paper the authors test the effects of a 5-day exposure to microplastic beads on the fighting behaviour of hermit crabs. They conduct a nicely designed experiment with a 2x2 factorial design, allowing them to investigate the relative effects of microplastic exposure on the behaviour of attackers and defenders, respectively. Overall it's a nice study but the paper needs a reasonable amount of work before it can be accepted for publication. There are several areas that lack clarity and I feel that the results require further interpretation. Please see below for my specific comments.

ABSTRACT

Lines 32-35: This sentence needs rewording to make your treatments and their effects clearer. I also suggest that you get rid of the quotation marks as they're not needed.

INTRODUCTION

Line 77: Why are hermit crabs important? Do you mean ecologically? If so, again, why are they ecologically important?

Lines 83-85: This sentence doesn't quite make sense, re-word.

Lines 86-88: Change the ending of this sentence to : - "on key survival behaviours reliant on such information processing."

Line 96: Make it clear that it is the quality of the defender's own shell you're talking about here.

In fact, here and throughout the manuscript you refer to the defender assessing the quality of the attacker's shell. However, defenders have very limited information about the attacker's shell due to the fact that they are withdrawn into their own shell for the entirety of the fight. Although it was traditionally thought that these shell fights were more of an amicable "negotiation", this has since been shown not to be the case. Thus it seems to me that defenders want to hold on to their own shell rather than be evicted which could lead to them either having no shell or inhabiting a shell they had no prior information on and thus could be worse than their current shell.

Line 119: Reference for this statement?

Line 120: Ecological consequences or fitness consequences? The latter is more in line with your previous statements.

Line 124: Why contestant role? Please elaborate. How do you expect the different roles to be affected?

Line 125: Surely a contest can only have one fight within it? Do you mean that crabs will be less willing to fight overall? Also, what do you mean by swaps? Do you just mean whether the attacker gets into the defender's shell after eviction? Does this not always happen?

METHODS

Line 143: While 4mm does technically equate to microplastic, it sounds huge in relation to the size of hermit crabs. Why did you use this size? Both in this and the previous paper when other papers such as that of Nanninga et al. 2020 use much smaller particles?

Line 148: Make it clear here that 20 crabs were allocated to each treatment tank.

Line 158-163: In order to make this clearer I would just refer to the larger crab as the attacker and the smaller crab as the defender. Otherwise it's a bit confusing.

Line 193: Again I don't understand entirely what you mean by 'shell swap' and moreover how you measured time to this? If by shell swap you mean the time taken for the attacker to get into the defender's shell upon evicting it, in my experience this happens immediately, is this not the case with your crabs?

Lines 196-205: Did you include relative weight difference in your models as a covariate in your analyses? RWD has previously been shown to impact various aspects of fighting behaviour, including in hermit crabs.

RESULTS

Line 220-112: If this interaction is significant, it's my understanding that you then cannot further assess the main effects of these variables as they are impacting each other. I would therefore just report the significant two-way interaction.

Line 225: I see from your R code that you performed post-hoc pairwise comparisons using Tukey's tests but you don't report the P values from these post-hoc tests here. I highly recommend that you do so that the reader can see which factor levels differ statistically from one another. The same applies for the duration of rapping.

DISCUSSION

Lines 252-253: I think you need to be careful here. Yes you looked at the effect of microplastics on information gathering but you didn't directly assess cognition. I think this needs to be toned down and elaborated on a bit more.

Lines 261-264: Could it be that exposure to the microplastic particles made them more risk-averse and therefore the defenders chose to stay in their shells rather than be evicted and thus possibly without a shell? A rubbish shell is after all better than no shell at all!

Line 267: It's interesting that the plastic-exposed attackers were worse at rapping but were just as likely to secure an eviction as control attackers. Why do you think this is? Why are the control defenders not using this information about the weak rappers and refusing eviction? Perhaps relate this to the findings of previous studies that have manipulated the physiological state of attackers.

Lines 270-271: How did you assess whether or not microplastic had been ingested? Did you look at the gut and gills?

Review form: Reviewer 2

Is the manuscript scientifically sound in its present form?

Yes

Are the interpretations and conclusions justified by the results?

Yes

Is the language acceptable?

Yes

Do you have any ethical concerns with this paper?

No

Have you any concerns about statistical analyses in this paper?

Yes

Recommendation?

Major revision is needed (please make suggestions in comments)

Comments to the Author(s)

The presence of microplastics is strongly increasing in the environment, especially in the aquatic ones, and it is thus crucial to assess their effects on species behaviour because they can indirectly affect the fitness and survival of animals. Hermit crabs are ideal organisms to study this topic because they live in the intertidal zone, often full of microplastics, and their entire life completely relies on shells. The paper is certainly interesting, showing how microplastics can alter shell assessment and fighting in hermits. Overall, the study is well presented and written. I have two

major comments to be addressed about the statistical analysis and the categorical variables used (please see the specific comments below).

Line 29: please do not use in the abstract the acronym.

Lines 130-131: is this beach free from plastics, so it can be assumed that the sampled hermits are not contaminated?

Lines 144-145: please explain here the meaning of 25 MP/L

Lines 175-176: this means that smaller crabs have shells too much heavy for them? Because this can affect their agonistic behaviour.

Line 202: a MANOVA would be more appropriate, considering that variables are not completely independent and that also the behaviour of attackers and defenders are correlated

Lines 190-191, 209-210: I am a little bit confused by these categorical variables and percentages, because they seem separated, but their sum is not 100%. So should we assume that a contest ended in a shell fight is a broad category including the subcategories contest ended with an eviction or a swap? This point is not completely clear, please better explain it.

Lines 277-282: after how many days of exposition to microplastics were these changes recorded in the species reported in the studies?

Line 288: have you considered to investigate the effect of nanoplastics too?

Decision letter (RSOS-211089.R0)

Dear Mr Cunningham

The Editors assigned to your paper RSOS-211089 "Animal contests and microplastics: Evidence of disrupted behaviour in hermit crabs *Pagurus bernhardus*" have now received comments from reviewers and would like you to revise the paper in accordance with the reviewer comments and any comments from the Editors. Please note this decision does not guarantee eventual acceptance.

Please submit your revised manuscript and required files (see below) no later than 21 days from today's (ie 24-Aug-2021) date. Note: the ScholarOne system will 'lock' if submission of the revision is attempted 21 or more days after the deadline. If you do not think you will be able to meet this deadline please contact the editorial office immediately.

Please note article processing charges apply to papers accepted for publication in Royal Society Open Science (<https://royalsocietypublishing.org/rsos/charges>). Charges will also apply to papers transferred to the journal from other Royal Society Publishing journals, as well as papers

submitted as part of our collaboration with the Royal Society of Chemistry (<https://royalsocietypublishing.org/rsos/chemistry>). Fee waivers are available but must be requested when you submit your revision (<https://royalsocietypublishing.org/rsos/waivers>).

on behalf of Professor Cinzia Chiandetti (Associate Editor) and Pete Smith (Subject Editor)
openscience@royalsociety.org

Associate Editor Comments to Author (Professor Cinzia Chiandetti):

Associate Editor: 1

Comments to the Author:

The manuscript represents a preliminary but well-designed and highly relevant observation in the field of environmental pollution and its effects on animal behaviour. The analysis of the results should be revised, and the reviewers have made specific suggestions as to which parts need to be corrected. Along with one reviewer, I agree that contests were used as a proxy for cognition without directly assessing the effects of the pollutant on it. Moreover, it is not entirely clear how it was determined that the crabs did not ingest microplastics during the experiment (lines 270-271). Also, I suggest better interpreting the results by avoiding restating them in the discussion (e.g. lines 248-252).

Reviewer comments to Author:

Reviewer: 1

Comments to the Author(s)

In this paper the authors test the effects of a 5-day exposure to microplastic beads on the fighting behaviour of hermit crabs. They conduct a nicely designed experiment with a 2x2 factorial design, allowing them to investigate the relative effects of microplastic exposure on the behaviour of attackers and defenders, respectively. Overall it's a nice study but the paper needs a reasonable amount of work before it can be accepted for publication. There are several areas that lack clarity and I feel that the results require further interpretation. Please see below for my specific comments.

ABSTRACT

Lines 32-35: This sentence needs rewording to make your treatments and their effects clearer. I also suggest that you get rid of the quotation marks as they're not needed.

INTRODUCTION

Line 77: Why are hermit crabs important? Do you mean ecologically? If so, again, why are they ecologically important?

Lines 83-85: This sentence doesn't quite make sense, re-word.

Lines 86-88: Change the ending of this sentence to : - "on key survival behaviours reliant on such information processing."

Line 96: Make it clear that it is the quality of the defender's own shell you're talking about here. In fact, here and throughout the manuscript you refer to the defender assessing the quality of the attacker's shell. However, defenders have very limited information about the attacker's shell due

to the fact that they are withdrawn into their own shell for the entirety of the fight. Although it was traditionally thought that these shell fights were more of an amicable “negotiation”, this has since been shown not to be the case. Thus it seems to me that defenders want to hold on to their own shell rather than be evicted which could lead to them either having no shell or inhabiting a shell they had no prior information on and thus could be worse than their current shell.

Line 119: Reference for this statement?

Line 120: Ecological consequences or fitness consequences? The latter is more in line with your previous statements.

Line 124: Why contestant role? Please elaborate. How do you expect the different roles to be affected?

Line 125: Surely a contest can only have one fight within it? Do you mean that crabs will be less willing to fight overall? Also, what do you mean by swaps? Do you just mean whether the attacker gets into the defender’s shell after eviction? Does this not always happen?

METHODS

Line 143: While 4mm does technically equate to microplastic, it sounds huge in relation to the size of hermit crabs. Why did you use this size? Both in this and the previous paper when other papers such as that of Nanninga et al. 2020 use much smaller particles?

Line 148: Make it clear here that 20 crabs were allocated to each treatment tank.

Line 158-163: In order to make this clearer I would just refer to the larger crab as the attacker and the smaller crab as the defender. Otherwise it's a bit confusing.

Line 193: Again I don't understand entirely what you mean by 'shell swap' and moreover how you measured time to this? If by shell swap you mean the time taken for the attacker to get into the defender's shell upon evicting it, in my experience this happens immediately, is this not the case with your crabs?

Lines 196-205: Did you include relative weight difference in your models as a covariate in your analyses? RWD has previously been shown to impact various aspects of fighting behaviour, including in hermit crabs.

RESULTS

Line 220-112: If this interaction is significant, it's my understanding that you then cannot further assess the main effects of these variables as they are impacting each other. I would therefore just report the significant two-way interaction.

Line 225: I see from your R code that you performed post-hoc pairwise comparisons using Tukey’s tests but you don’t report the P values from these post-hoc tests here. I highly recommend that you do so that the reader can see which factor levels differ statistically from one another. The same applies for the duration of rapping.

DISCUSSION

Lines 252-253: I think you need to be careful here. Yes you looked at the effect of microplastics on information gathering but you didn't directly assess cognition. I think this needs to be toned down and elaborated on a bit more.

Lines 261-264: Could it be that exposure to the microplastic particles made them more risk-averse and therefore the defenders chose to stay in their shells rather than be evicted and thus possibly without a shell? A rubbish shell is after all better than no shell at all!

Line 267: It's interesting that the plastic-exposed attackers were worse at rapping but were just as likely to secure an eviction as control attackers. Why do you think this is? Why are the control defenders not using this information about the weak rappers and refusing eviction? Perhaps relate this to the findings of previous studies that have manipulated the physiological state of attackers.

Lines 270-271: How did you assess whether or not microplastic had been ingested? Did you look at the gut and gills?

Reviewer: 2

Comments to the Author(s)

The presence of microplastics is strongly increasing in the environment, especially in the aquatic ones, and it is thus crucial to assess their effects on species behaviour because they can indirectly affect the fitness and survival of animals. Hermit crabs are ideal organisms to study this topic because they live in the intertidal zone, often full of microplastics, and their entire life completely relies on shells. The paper is certainly interesting, showing how microplastics can alter shell assessment and fighting in hermits. Overall, the study is well presented and written. I have two major comments to be addressed about the statistical analysis and the categorical variables used (please see the specific comments below).

Line 29: please do not use in the abstract the acronym.

Lines 130-131: is this beach free from plastics, so it can be assumed that the sampled hermits are not contaminated?

Lines 144-145: please explain here the meaning of 25 MP/L

Lines 175-176: this means that smaller crabs have shells too much heavy for them? Because this can affect their agonistic behaviour.

Line 202: a MANOVA would be more appropriate, considering that variables are not completely independent and that also the behaviour of attackers and defenders are correlated

Lines 190-191, 209-210: I am a little bit confused by these categorical variables and percentages, because they seem separated, but their sum is not 100%. So should we assume that a contest ended in a shell fight is a broad category including the subcategories contest ended with an eviction or a swap? This point is not completely clear, please better explain it.

Lines 277-282: after how many days of exposition to microplastics were these changes recorded in the species reported in the studies?

Line 288: have you considered to investigate the effect of nanoplastics too?

===PREPARING YOUR MANUSCRIPT===

If you have been asked to revise the written English in your submission as a condition of publication, you must do so, and you are expected to provide evidence that you have received language editing support. The journal would prefer that you use a professional language editing

service and provide a certificate of editing, but a signed letter from a colleague who is a native speaker of English is acceptable. Note the journal has arranged a number of discounts for authors using professional language editing services (<https://royalsociety.org/journals/authors/benefits/language-editing/>).

===PREPARING YOUR REVISION IN SCHOLARONE===

-- If you have uploaded ESM files, please ensure you follow the guidance at <https://royalsociety.org/journals/authors/author-guidelines/#supplementary-material> to include a suitable title and informative caption. An example of appropriate titling and captioning

may be found at https://figshare.com/articles/Table_S2_from_Is_there_a_trade-off_between_peak_performance_and_performance_breadth_across_temperatures_for_aerobic_sc_ope_in_teleost_fishes_/3843624.

Author's Response to Decision Letter for (RSOS-211089.R0)

See Appendix A.

Decision letter (RSOS-211089.R1)

Dear Mr Cunningham,

It is a pleasure to accept your manuscript entitled "Animal contests and microplastics: Evidence of disrupted behaviour in hermit crabs *Pagurus bernhardus*" in its current form for publication in Royal Society Open Science. The comments of the reviewer(s) who reviewed your manuscript are included at the foot of this letter.

on behalf of Professor Cinzia Chiandetti (Associate Editor) and Pete Smith (Subject Editor)
openscience@royalsociety.org

Appendix A

RSOS-211089 “Animal contests and microplastics: Evidence of disrupted behaviour in hermit crabs
Pagurus bernhardus”

Reviewer responses

Associate Editor:

The manuscript represents a preliminary but well-designed and highly relevant observation in the field of environmental pollution and its effects on animal behaviour. The analysis of the results should be revised, and the reviewers have made specific suggestions as to which parts need to be corrected. Along with one reviewer, I agree that contests were used as a proxy for cognition without directly assessing the effects of the pollutant on it. Moreover, it is not entirely clear how it was determined that the crabs did not ingest microplastics during the experiment (lines 270-271). Also, I suggest better interpreting the results by avoiding restating them in the discussion (e.g. lines 248-252).

We thank the Editor for their feedback on our study, and agree that contests are only used as a proxy for cognition in hermit crabs. We have instead referred to microplastics affecting ‘decision making’ in hermit crabs as opposed to ‘cognition’ throughout the manuscript. In addition, we have added to the discussion that no microplastic ingestion was possible throughout the experiments as the particles floated visibly on the surface of the tanks at all times and were not accessible by the hermits (See L283). Furthermore, we have amended the wording that summarises our findings in the discussion and placed them in the context of a relevant interpretation.

Reviewer 1:

In this paper the authors test the effects of a 5-day exposure to microplastic beads on the fighting behaviour of hermit crabs. They conduct a nicely designed experiment with a 2x2 factorial design, allowing them to investigate the relative effects of microplastic exposure on the behaviour of attackers and defenders, respectively. Overall it’s a nice study but the paper needs a reasonable amount of work before it can be accepted for publication. There are several areas that lack clarity and I feel that the results require further interpretation. Please see below for my specific comments.

We thank the reviewer for their constructive feedback on our paper. We have addressed the specific comments below and have added greater clarity throughout the paper.

ABSTRACT

Lines 32-35: This sentence needs rewording to make your treatments and their effects clearer. I also suggest that you get rid of the quotation marks as they’re not needed.

We have reworded to make this clearer:

L32-35 “Significantly higher raps per bout were needed to evict microplastic-treated defenders when attackers were pre-exposed to control conditions (i.e. no plastic). Also, significantly longer durations of rapping bouts were needed to evict control-treated defenders when attackers were pre-exposed to microplastics.”

Quotation marks also removed.

INTRODUCTION

Line 77: Why are hermit crabs important? Do you mean ecologically? If so, again, why are they ecologically important?

We have simplified this by removing the word important:

L77 “*Hermit crabs, a widespread marine crustacean*”

Lines 83-85: This sentence doesn't quite make sense, re-word.

Reworded to:

L83-85 “*Hermit crabs gather information by investigating a new shell, which allows them to decide whether it is of a higher or lower quality than the shell they currently inhabit (Arnott & Elwood, 2007; Briffa & Elwood, 2004; Elwood & Neil, 1992).*”

Lines 86-88: Change the ending of this sentence to : - "on key survival behaviours reliant on such information processing."

Changed. See L88

Line 96: Make it clear that it is the quality of the defender's own shell you're talking about here. In fact, here and throughout the manuscript you refer to the defender assessing the quality of the attacker's shell. However, defenders have very limited information about the attacker's shell due to the fact that they are withdrawn into their own shell for the entirety of the fight. Although it was traditionally thought that these shell fights were more of an amicable “negotiation”, this has since been shown not to be the case. Thus it seems to me that defenders want to hold on to their own shell rather than be evicted which could lead to them either having no shell or inhabiting a shell they had no prior information on and thus could be worse than their current shell.

We have changed the sentence to make it clear that the defender assesses contest costs versus the resource value of its own shell.

L96 “*determine whether contest costs (i.e. energy, injury, time) outweigh the quality of their own shell (i.e. resource value; Arnott & Elwood, 2008).*”

We have also made this clearer throughout the manuscript.

Line 119: Reference for this statement?

Reference added and statement changed to:

L119-121 “*Therefore, any impairment in this behaviour associated with microplastic exposure could have major consequences for environmental fitness, as lower quality shells can reduce growth, fecundity, and survival (Lancaster, 1990).*”

Line 120: Ecological consequences or fitness consequences? The latter is more in line with your previous statements.

Changed to fitness consequences. See previous comment. L119-121

Line 124: Why contestant role? Please elaborate. How do you expect the different roles to be affected?

We have added a section to elaborate on the importance of examining contestant role:

L125-129 *“The latter is important because the roles of attacker and defender are very different and distinctive in hermit crab contests, with attackers being highly active and using bouts of shell rapping to try to evict the defender who withdraws inside their shell and attempt to resist eviction. Given this, it is therefore important to investigate if microplastic exposure influences one contestant role more than the other.”*

Line 125: Surely a contest can only have one fight within it? Do you mean that crabs will be less willing to fight overall? Also, what do you mean by swaps? Do you just mean whether the attacker gets into the defender’s shell after eviction? Does this not always happen?

Here we refer to the number of contests overall and how they may be affected by microplastic exposure, rather than the number of contests each individual will have. Each crab fought once. In terms of swaps, we observed many instances where the attacker would evict the defender and then swap shells; however, the attacker sometimes returns to its original shell – we did not classify this as a shell swap. Therefore one of our categorical variables was shell swap (Yes/No). We have adjusted this sentence to make it clearer:

L129-132 *“Additionally, we predicted that the overall outcome of hermit crab contests (i.e. the number of shell fights, evictions, and swaps overall) would be affected by the presence/absence of microplastic pollution, and/or the contestant role.”*

METHODS

Line 143: While 4mm does technically equate to microplastic, it sounds huge in relation to the size of hermit crabs. Why did you use this size? Both in this and the previous paper when other papers such as that of Nanninga et al. 2020 use much smaller particles?

We agree that smaller particles would be more realistic; however, we wanted to keep the methodology the same as Crump et al. 2020 paper. By using larger particles we could also ensure that the hermits were not ingesting the microplastics and therefore any effects were as a result of exposure.

Line 148: Make it clear here that 20 crabs were allocated to each treatment tank.

Changed:

L154 *“20 crabs were allocated to each treatment tank using a random number generator and exposed for five-days”*

Line 158-163: In order to make this clearer I would just refer to the larger crab as the attacker and the smaller crab as the defender. Otherwise it's a bit confusing.

Agreed. We have changes this to read as the following:

L166-167: *“3) the attacker from the MP and the defender from the C (MPA); and 4) the attacker from the C and the defender from the MP (MPD).”*

Line 193: Again I don't understand entirely what you mean by 'shell swap' and moreover how you measured time to this? If by shell swap you mean the time taken for the attacker to get into the

defender's shell upon evicting it, in my experience this happens immediately, is this not the case with your crabs?

The shell swap is the time taken for the attacker to get into the defender's shell and stay there. We previously observed multiple swaps before the attacker decides which shell to take. However, no significant difference in the categorical variables was found among the treatments anyway. This has now been changed to the following:

L200 “and the time taken for the attacker to swap and remain in the defender's shell (s).”

Lines 196-205: Did you include relative weight difference in your models as a covariate in your analyses? RWD has previously been shown to impact various aspects of fighting behaviour, including in hermit crabs.

We thank the reviewer for their comment. We have now added to the methodology that we calculated the RWD of fighting pairs and added this a covariate in our analysis.

L181-182 “In addition, the relative weight difference (RWD) of the pair was calculated by the following equation; $RWD = 1 - (\text{small crab weight}/\text{large crab weight})$.”

This has also been added to L201 where the numerical variables are described.

In this instance, there was no effect of RWD on our study, and the results have been added to the supplementary material with the other non-significant variables:

L239-240 “Furthermore, all of the other variables measured with numerical data were statistically non-significant (see Supplementary Table. 1).”

RESULTS

Line 220-112: If this interaction is significant, it's my understanding that you then cannot further assess the main effects of these variables as they are impacting each other. I would therefore just report the significant two-way interaction.

We have changed this to report only the interaction effect:

L229-230 “However, a significant microplastic exposure \times contestant role interaction was found for the ‘number of raps within a bout’ ($F_1 = 5.88, P = 0.02$).”

Line 225: I see from your R code that you performed post-hoc pairwise comparisons using Tukey's tests but you don't report the P values from these post-hoc tests here. I highly recommend that you do so that the reader can see which factor levels differ statistically from one another. The same applies for the duration of rapping.

We did not report the Tukey's test results as they were all non-significant despite the significant interactions effects. We have now added this to the statistical methods section:

L212-214 “In the case of a significant interaction effect, a post hoc Tukey's pairwise comparison test was used to highlight the factor levels that differed significantly from each other.”

And as a comment in the results section:

L230-234 “Despite the post hoc analysis reporting that no individual factor level differed significantly from each other, the significant interaction was driven by a higher average increase in

number of raps needed by attackers from control conditions to evict microplastic-exposed defenders from their shells (MPD; Fig.1a).”

L236-239 *“Again, despite a non-significant post hoc analysis, the highly significant interaction was driven by longer average durations of rapping bouts needed by attackers from microplastic conditions to evict control-exposed defenders from their shell (MPA; Fig.1b).”*

DISCUSSION

Lines 252-253: I think you need to be careful here. Yes you looked at the effect of microplastics on information gathering but you didn't directly assess cognition. I think this needs to be toned down and elaborated on a bit more.

Agreed. We have changed this statement to read:

L260-262 *“We show that microplastics affect information gathering during these behaviours, however, further research is needed to see if microplastics directly affect hermit crab cognition.”*

We have also removed this statement throughout the manuscript. See L271-272.

Lines 261-264: Could it be that exposure to the microplastic particles made them more risk-averse and therefore the defenders chose to stay in their shells rather than be evicted and thus possibly without a shell? A rubbish shell is after all better than no shell at all!

Thanks for this great suggestion. We agree, and have added this as a possible explanation to the paragraph.

L269-271 *“It may also be the case that the defenders exposed to microplastics became more risk-averse and decided to stay in their shells for longer despite facing an inevitable eviction.”*

Line 267: It's interesting that the plastic-exposed attackers were worse at rapping but were just as likely to secure an eviction as control attackers. Why do you think this is? Why are the control defenders not using this information about the weak rappers and refusing eviction? Perhaps relate this to the findings of previous studies that have manipulated the physiological state of attackers.

We think that the microplastic leachate may have affected the physiology of the attackers and reduced their rapping strength. However, it is unusual that the affected contest mechanisms did not change the outcome of the contests overall. Unfortunately, we did not test the physiology of the animals to investigate this further. We have added further interpretation of the reduced attacking ability and related this to previous studies on the physiology of hermits during contests.

L275-283 *“The mechanisms behind the reduced attacking ability of microplastic-exposed attackers when facing control defenders is most likely a result of the attackers reduced physiological condition (Briffa & Elwood, 2002), as fighting ability largely depends on physiological condition of the animal (Doake & Elwood, 2011). Previous studies have shown how the rapping vigour of an attacker can be reduced as lactate levels increase throughout a contest (Briffa & Elwood, 2001). Here, we highlighted longer durations of rapping bouts without having increased number of raps, which suggests that the rapping behaviour was either slower or weaker. It is therefore possible that the five-day exposure to microplastics affected the competitive ability of the hermit crabs.”*

Lines 270-271: How did you assess whether or not microplastic had been ingested? Did you look at the gut and gills?

The microplastic particles were floating on the surface of the water and were out of reach of the hermit crabs throughout the experiments. We have now added this to the section:

L283-284 “As the microplastics floated on the surface of the water and no microplastic ingestion was observed by the hermit crabs during the experiment”

Reviewer 2:

The presence of microplastics is strongly increasing in the environment, especially in the aquatic ones, and it is thus crucial to assess their effects on species behaviour because they can indirectly affect the fitness and survival of animals. Hermit crabs are ideal organisms to study this topic because they live in the intertidal zone, often full of microplastics, and their entire life completely relies on shells. The paper is certainly interesting, showing how microplastics can alter shell assessment and fighting in hermits. Overall, the study is well presented and written. I have two major comments to be addressed about the statistical analysis and the categorical variables used (please see the specific comments below).

We thank the reviewer for their positive and constructive feedback. We have added further clarity on the categorical variables to make this more easily understood. We also thank the reviewer for their suggestion of using a MANOVA. We have investigated this approach. However, given we only have a limited number of dependent variables (with little risk of Type 1 error) we prefer to use the two-factor ANOVA approach as it is appropriate for our data and will be familiar to a broad audience of readers, also enabling us to present the data clearly in our chosen figures.

Line 29: please do not use in the abstract the acronym.

Acronym replaced with “microplastics/litre”

Lines 130-131: is this beach free from plastics, so it can be assumed that the sampled hermits are not contaminated?

We did not test the beach for microplastic contamination, however we intended to avoid this by not taking sediment from the site where microplastic pollution is known to accumulate (Cunningham & Sigwart, 2019). Sites in close proximity to Ballywalter have also shown low levels of microplastic contamination in the past (~1 microplastic/litre; Green et al. 2018). We also maintained our animals on sand-filtered seawater to ensure minimal microplastic contamination from the water. We are confident that the hermit crabs were not contaminated.

Lines 144-145: please explain here the meaning of 25 MP/L

Meaning of MP/L added to L151.

Lines 175-176: this means that smaller crabs have shells too much heavy for them? Because this can affect their agonistic behaviour.

Yes, the smaller crabs were given the shell best suited to the larger crab. Therefore the defender’s shell was heavier. This asymmetry is needed to encourage the attacker to engage in a shell fight and has been used in previous hermit crab papers (Arnott & Elwood, 2007).

Line 202: a MANOVA would be more appropriate, considering that variables are not completely independent and that also the behaviour of attackers and defenders are correlated

We agree that the behaviour of attackers and defenders within a contest is not independent which is why we use the contest dyad as the unit of observation. We also thank the reviewer for their suggestion of using a MANOVA. We have investigated this approach. However, given we only have a limited number of behavioural dependent variables (with little risk of Type 1 error) we prefer to use the two-factor ANOVA approach as it is appropriate for our data and will be familiar to a broad audience of readers, also enabling us to present the data clearly in our chosen figures. In this regard, we designed our study on two previous hermit crab contest studies that utilised a two-factor ANOVA approach to describe the effects of two treatment groups on behavioural variables (Arnott & Elwood 2007, 2010). Our study design and analysis were therefore chosen prior to the data collection as it was considered to be the most appropriate analysis at the time. As described above, this design also allows us to present our data clearly in the figures and be more easily understood.

Lines 190-191, 209-210: I am a little bit confused by these categorical variables and percentages, because they seem separated, but their sum is not 100%. So should we assume that a contest ended in a shell fight is a broad category including the subcategories contest ended with an eviction or a swap? This point is not completely clear, please better explain it.

We have now removed the term ‘contests that ended in a shell fight etc’ as this should actually be ‘contests that included a shell fight etc’. The contests did not only include one of these outcomes, but sometimes multiple outcomes. Therefore the percentages reflect the overall number of contests that included a shell fight, eviction, or shell swap. As the percentages for each outcome were statistically the same, it shows that when there was a shell fight, there was also an eviction and a swap. This has been made clearer in the following places:

L196-197 “The measured categorical variables (Yes/No) were: contests that included a shell fight, contests that included an eviction, and contests that included a shell swap.”

L218-219 “Among all treatments overall, 67 – 77% of contests included shell fights, 67 – 73% included shell evictions, and 57 – 70% included shell swaps”

L224-226 “The number and percentage of contests that contained a shell fight, shell eviction, or shell swap overall for each of the four treatments; control (C), microplastic (MP), microplastic attacker (MPA), and microplastic defender (MPD).”

Lines 277-282: after how many days of exposition to microplastics were these changes recorded in the species reported in the studies?

The leachate exposure duration for each of the mentioned studies were as follows: 24 hours (Ke et al. 2019), 4 days (Li et al. 2016), and 5 days (Green et al. 2021). These values have now been added to the discussion. See L292, 295, 299.

Line 288: have you considered to investigate the effect of nanoplastics too?

Thanks for this great suggestion. We have not as yet, but will consider this for future studies. This has now been added to the discussion. See L303.